# Enhancing Anti-PD-1 Immunotherapy by Targeting MDSCs via Hepatic Arterial Infusion in Breast Cancer Liver Metastases

**DOI:** 10.3390/cancers16213711

**Published:** 2024-11-03

**Authors:** Minhyung Kim, Colin A. Powers, Daniel T. Fisher, Amy W. Ku, Nickolay Neznanov, Alfiya F. Safina, Jianmin Wang, Avishekh Gautam, Siddharth Balachandran, Anuradha Krishnamurthy, Katerina V. Gurova, Sharon S. Evans, Andrei V. Gudkov, Joseph J. Skitzki

**Affiliations:** 1Department of Surgical Oncology, Roswell Park Comprehensive Cancer Center, Buffalo, NY 14263, USA; powersca@upmc.edu (C.A.P.); daniel.fisher@roswellpark.org (D.T.F.); ak4448@cumc.columbia.edu (A.W.K.); 2Department of Immunology, Roswell Park Comprehensive Cancer Center, Buffalo, NY 14263, USA; sharon.evans@roswellpark.org; 3Department of Cell Stress Biology, Roswell Park Comprehensive Cancer Center, Buffalo, NY 14263, USA; lubovnez956@gmail.com (N.N.); alfiya_safina@yahoo.com (A.F.S.); katerina.gurova@roswellpark.org (K.V.G.); andrei.gudkov@roswellpark.org (A.V.G.); 4Department of Biostatistics & Bioinformatics, Roswell Park Comprehensive Cancer Center, Buffalo, NY 14263, USA; jianmin.wang@roswellpark.org; 5Blood Cell Development and Function Program, Fox Chase Cancer Center, Philadelphia, PA 19111, USA; avishekh.gautam@fccc.edu (A.G.); siddharth.balachandran@fccc.edu (S.B.); 6Department of Medicine, Roswell Park Comprehensive Cancer Center, Buffalo, NY 14263, USA; anuradha.krishnamurthy@roswellpark.org

**Keywords:** hepatic arterial infusion (HAI), immunotherapy, liver tumor, myeloid-derived suppressor cells (MDSCs), Z-DNA

## Abstract

Various cancer therapies are often ineffective for advanced metastatic liver disease. While immunotherapy can be effective in some cases, it does not provide significant benefits for metastatic cancer in the liver. CBL0137 is an experimental drug that helps form a special type of DNA called Z-DNA, which may help boost the body’s immune response against tumors. We investigated how CBL0137 affects metastatic liver tumor models from colon (CT26) and breast (4T1) cancers focusing on how it triggers immune responses. The results showed that CBL0137 hepatic arterial infusion (HAI) enhanced anti-tumor effects by depleting immune-suppressing cells while preserving effector T cells. Combining CBL0137 HAI with conventional immunotherapy improved survival in 4T1 tumors but not in CT26 tumors, highlighting the importance of targeting specific immune cell populations for effective treatment.

## 1. Introduction

The liver is the most common site for solid tumor metastases and the dismal 5-year survival rate below 10% underlies the urgent need for improved therapies [1,2,3,4,5]. In particular, colon and breast liver metastases remain problematic with ~20–25% of colon cancer patients having liver metastases at diagnosis [6,7], and the liver represents the third most common metastatic site for breast cancer [8]. Surgery remains the mainstay of treatment for these patients as it offers the best chance of complete tumor removal and has been shown to improve overall survival [9,10,11,12]. Unfortunately, the majority of patients are not eligible candidates for surgery due to unresectable lesions, inadequate estimated post-surgical liver volume, and/or concurrent hepatic pathology. In non-operable patients, options such as systemic chemotherapy, chemoembolization, hyperthermic or radiofrequency ablative therapies, and local injections of chemotherapy into identifiable tumors are employed for survival prolongation with an extremely low probability of cure [13,14,15].

Current systemic chemotherapy regimens in the setting of metastatic liver disease are often limited by systemic toxicity [16,17,18]. As an alternative to systemic treatments, chemotherapy can be administered regionally to the liver via hepatic arterial infusion (HAI) [19,20]. HAI is uniquely primed for the treatment of metastatic cancers within the liver as liver metastases derive nutrients solely from the hepatic artery [21,22] and direct administration of compounds via a regional artery creates a first-pass effect leading to higher concentrations of the drug reaching the tumor bed and limiting systemic absorption or metabolism [23]. Floxuridine (FUDR) is the most commonly used agent for HAI as it is quickly metabolized to 5-fluorouracil (5-FU) at the level of the tumor and achieves a high first-pass effect (94–99%) that limits systemic toxicity. However, despite the theoretical benefit of HAI, survival data have only been marginally impacted in select patients [24,25].

Our group has identified the small molecule CBL0137 as a member of the novel class of anti-cancer agents known as curaxins [26]. CBL0137 is currently in clinical trials for the treatment of advanced melanoma, sarcoma, and hematological malignancies (NCT03727789 and NCT02931110). CBL0137 binds DNA without causing DNA damage. Rather, CBL0137 destabilizes DNA and histone interactions resulting from DNA segment elongation and produces nucleosome disassembly with multiple downstream functional consequences of chromatin decondensation in tumor cells (Figure 1) [26]. A measurable outcome is the elaboration of Z-DNA conformational changes and functional inactivation of the histone chaperone named Facilitates Chromatin Transcription (FACT). FACT is a heterodimeric complex composed of structure-specific recognition protein 1 (SSRP1) and a suppressor of ty-16 (SPT16) and is involved in transcription, replication, and DNA repair [27,28,29]. FACT overexpression in multiple tumor types (e.g., non-small-cell lung cancer (NSCLC), pancreatic cancer, breast cancer, and neuroblastoma) correlates with poor prognosis [30,31,32]. CBL0137-induced nucleosome disassembly in vitro and in vivo causes FACT to bind partially disassembled nucleosomes with high affinity, an event known as chromatin trapping (C-trapping) [33]. C-trapping reduces the availability of FACT, decreasing the FACT-dependent transcription of target genes (e.g., NF-kB, HSF1, HIF1α, MYC) that leads to casein kinase 2 (CK2)-dependent activation of tumor suppressor p53 [26]. Therefore, the FACT subunit SSRP1, C-trapping, and Z-DNA production have emerged as reliable readouts for CBL0137 function. Since CBL0137 is markedly water-soluble and cell-permeable and has a high affinity for DNA with prolonged interaction even after removal of the drug from the cell culture medium or circulation [34,35], it is a promising candidate for HAI and could be superior to the currently employed 5-FU-based therapeutics.

In this study, we compared the efficacy of conventional 5-FU HAI with CBL0137 in syngeneic murine models of colorectal (CT26) and metastatic breast (4T1) tumors in the liver. Our current study demonstrates the direct anti-tumor efficacy of CBL0137 HAI and a selective modification of the tumor microenvironment by depleting myeloid-derived suppressor cells (MDSCs) and enhancing effector immune populations. Furthermore, a combination of CBL0137 with programmed cell death (PD)-1 blockade therapy improves efficacy, thus supporting a novel paradigm for the treatment of liver metastases.

## 2. Materials and Methods

### 2.1. Cell Lines

4T1-Luc, 4T1 NF-κB-Luc, CT26-Luc, and CT26 NF-κB-Luc cells were cultured in RPMI media (Corning Inc., Corning, NY, USA) containing 10% FBS, 0.1% Penicillin/Streptomycin, and 0.05% 2-mercaptoethanol, supplemented with 10% L-glutamine (Gibco, Gaithersburg, MD, USA). 4T1-Luc and CT26-Luc cell lines were generated by the authors using a well-established method [36]. 4T1 and CT26 NF-κB-Luc cell lines were reporter cell lines with luciferase under the transcriptional control of NF-κB.

### 2.2. Chemicals and Reagents

CBL0137 (lot 10-106-88-30, Aptuit, Greenwich, CT, USA) in 5% dextrose was generously provided by Incuron LLC (Buffalo, NY, USA). 5FU was purchased from Sigma-Aldrich (St. Louis, MO, USA) and luciferin was purchased from Promega (Madison, WI, USA). In vivo MAb anti-mouse PD-1 (CD279; Clone RMP1-14) was purchased from BioXcell (Lebanon, NH, USA), and 200 μg of the antibody was injected 3 times a week intraperitoneally.

### 2.3. Mice

Female BALB/c mice aged 7–8 weeks were purchased from Charles River (Kingston, NY, USA). All mice were fed a standard laboratory diet and housed under standard lighting and housing conditions.

### 2.4. Tumor Establishment

A hemisplenectomy technique was used to model liver metastasis [37]. Briefly, mice were induced with 4% isoflurane and kept in the appropriate anesthetic plane with 1.5% isoflurane via a nose cone. After removing the hair over the proposed incision site and sterilizing the area with iodine and alcohol, the mouse was draped, a flank incision was carried down into the abdominal cavity, and the spleen was gently extracted. Two 5-0 silk ties were placed in the body of the spleen and the spleen was transected between the two ties, and minor bleeding was controlled by electrocauterization. The superior pole was placed back into the abdomen. In total, 2 × 10^5^ 4T1-Luc or CT26- Luc cells in 200 µL sterile PBS were injected into the inferior pole. The area was irrigated, and the inferior pole was ligated with a 5-0 silk tie and removed with cautery. A subcutaneous tumor model was generated after injecting the tumor cells (100 μL) subcutaneously using a 30 G needle in the left flank.

### 2.5. HAI Procedure

The superior pancreaticoduodenal artery (SPDA) model was adapted for this study [38]. Briefly, the hepatic artery (HA) and SPDA were obtained under microscopy, and a 6-0 silk tie was placed around the distal SPDA, and then a vascular clip was applied to the celiac artery (CA). An arteriotomy was created using microscissors on the SPDA, and a microcatheter was advanced near the opening of the hepatic artery without flow disturbance of the CA (Figure 2). The vascular clip was released after securing the microcatheter. The ischemia time was less than 5 min. After cannulation, the drug or vehicle was delivered by a peristaltic pump (Cole-Parmer Instrument Co., Vernon Hills, IL, USA) with 40 µL/min of a flow rate for 15 min at room temperature. The silk tie was used to fully ligate the proximal portion of the SPDA to prevent bleeding after infusion. All mice were recovered on a warming blanket and injected subcutaneously with Ethiqa XR (3.25 mg/kg body weight, Fidelis Animal Health, North Brunswick, NJ, USA) for pain control and 0.5 mL of normal saline for hydration.

### 2.6. In Vitro Cytotoxicity Assay

4T1 tumor cells were plated at a rate of 5 × 10^5^ in a 96-well plate in a total of 100 µL of culture media and allowed to settle for 24 h at 37 degrees. After 24 h, the culture media was aspirated from each well, and the cells were treated with either 100 µL of PBS or 100 µL of the drug (CBL0137 or 5FU) at concentrations of 0.5 mg/mL, 1 mg/mL, and 2 mg/mL. After 15 min, the drug mixture was completely aspirated out of the wells, and 100 µL of fresh, pre-warmed culture media was added back into the wells. After allowing 3 h for the cells to return to equilibrium, viability was measured using the CytoSelect MTT Cell Proliferation Assay (Cell Biolabs, Inc., San Diego, CA, USA) [39,40].

### 2.7. Assessment of NF-κB Activity and Tumor Responses

4T1 and CT26 NF-κB-Luc cell lines were used for the evaluation of NF-κB inhibition with CBL0137. LPS (2.5 mg/kg, Sigma-Aldrich) was injected intraperitoneally into normal saline (200 μL of total volume) to activate NF-κB transcription, and luminescent signals were measured 30 min after LPS injection and 4 h after HAI treatment. For tumor response studies, 4T1-Luc or CT26-Luc liver metastasis-bearing mice were infused either with the vehicle, 1.0 mg of 5FU, or 0.5 mg of CBL0137 by HAI or systemic routes (tail vein) 7 or 10 days after tumor inoculation, respectively. Tumor responses were checked 0, 2, and 5 days (or 4 times for 10 days) after treatment using in vivo luciferase bioluminescence measurements. Animals were injected with 0.15 mg/g of D-luciferin (GoldBio, St Louis, MO, USA) intraperitoneally 5 min before the measurement. Values are reported as means ± standard errors for each experimental group.

### 2.8. Assessment of Safety and Toxicity

The body weights of mice were measured 2, 3, 7, 11, and 14 days after the HAI procedure using a digital scale (AWS Inc., Saturn Court Norcross, GA, USA). Aspartate aminotransferase (AST), alanine aminotransferase (ALT), total bilirubin (Tbili), and amylase levels were measured 1, 2, and 3 days after HAI as markers of liver and pancreas toxicity [41]. All blood chemistry levels were measured in plasma using VITROS 5.1 FS (Ortho Clinical Diagnostics, Inc., Rochester, NY, USA).

### 2.9. Western Immunoblotting

Total soluble cellular protein extracts were prepared in a RIPA buffer [150 mmol/L NaCl, 1% SDS, 10 mmol/L Tris (pH 8.0), 1% sodium deoxycholate, 1% NP-40] containing a protease inhibitor cocktail (Sigma-Aldrich). The anti-SSRP1 antibody (clone; D-7) was from Santa Cruz Biotechnology (Dallas, TX, USA), and a rabbit anti-actin antibody was used to control for protein loading (Sigma-Aldrich). This technique has been previously reported [33]. Image quantification was performed using the ImageJ program (NIH) [42].

### 2.10. Isolation of Hepatic and Splenic Leukocytes

The liver tissue was collected and finely minced using a razor blade. The minced tissue was then placed into 10 mL of an enzyme solution containing 1 mg/mL collagenase type IV (Worthington Biochemical, Lakewood, NJ, USA) and 10 U/mL of DNase (Sigma-Aldrich) with shaking for 1 h at 37 °C. The tissue was filtered through a 70 μm cell strainer, flushed with 10 mL PBS, and centrifuged for 5 min at 380 g. After the elimination of red blood cells by hypotonic lysis, leukocytes were isolated by centrifugation at 890 g for 15 min at 20 °C using 3 mL Lymphoprep (Stemcell Technologies, Vancouver, Canada) and 4 mL of cell suspension in DPBS. The spleen was pushed and ground through a 70 µm cell strainer and washed with DPBS to create a cell suspension of splenocytes. The cells were centrifuged at 650 g at 4 °C for 5 min and then incubated for 5 min in 5 mL of the RBC lysis buffer at room temperature.

### 2.11. Flow Cytometry

Spleens were mechanically disrupted and directly passed through a 70 μm nylon cell strainer (Alkali Scientific, Pompano Beach, FL, USA) followed by hypotonic lysis of red blood cells (ACK lysis buffer; Gibco). Liver and splenic leukocytes were stained for extracellular and intracellular markers using antibodies from BD Bioscience, such as CD45 BUV395 30-F11 1:200, CD3 AF700 17A2 1:100, CD4 PerCp RM4.5 1:100, CD8 AF488 53-6.7 1:100, CD25 APC PC61 1:100, CD11b PE-Cy7 M1/70 3:2000, Gr1 BV421 RB6-8C5 1:100, Ly6g BUV395 1A8 1:100, Ly6c APC AL-21 1:100, PD1 BUV737 RMP1-30 1:100, Foxp3 PE MF23 1:100, IFN-γ PE-CF-594 XMG1.2 1:200, IL4 APC 11B11 1:100, and IL17 BV421 TC11-18H10 1:100. All data were collected on an LSR Fortessa flow cytometer (BD Biosciences, San Jose, CA, USA) and analyzed with WinList 9.0 software (Verity Software House, Topsham, ME, USA).

### 2.12. Detection of Apoptotic Cells, Gr-1^+^, CD8^+^ Cells, and Z-DNA

Non-sequential 9 µm thick sections were prepared from snap-frozen liver tissues. TUNEL staining was performed using the ApopTag Kit (Millipore Corporation, Burlington, MA, USA). To detect CD8^+^ T cells, liver sections were stained with Gr-1, CD8, or isotype-matched control antibodies. Anti-Z DNA antibodies were purchased from Abcam. A fluorescence microscope (Olympus, Tokyo, Japan) with a SPOT RT color camera (Diagnostic Instruments, Inc., Sterling Heights, MI, USA) was used to obtain photographs of the tissue. The data represent the average TUNEL-positive, Gr-1-positive, or CD8-positive cells in tumors per mouse.

### 2.13. T Cell Suppression Assay

To evaluate T cell suppression during continuous exposure to MDSCs, splenic and hepatic CD11b^+^ cells were purified from 4T1 liver metastasis-bearing BALB/c mice using anti-CD11b^+^ magnetic beads (Miltenyi Biotec, San Diego, CA, USA) as per the provided manual. Isolated populations were confirmed for their biomarkers with flow cytometry. Isolated cells were combined at the indicated ratios with 10^6^ CFSE-labeled (ThermoFisher Scientific, Waltham, MA, USA) splenocytes from tumor-free BALB/c mice in a 24-well plate and cultured for 96 h with anti-CD3/CD28 antibody-conjugated beads (1 µL per 100 µL culture; ThermoFisher Scientific) and IL-2 (30 U/mL; Peprotech, Rocky Hill, NJ, USA) in 2 mL of MLM cell culture media which was composed of 500 mL RPMI, 5.5 mL Glutamax, 1 mL Pen/Strep, 570 µL 2-mercaptoethanol, 5.5 mL Na Pyruvate, 10 mL non-essential amino acids, 10 mL HEPES, and 50 mL of FBS. T cell proliferation was measured based on CFSE dilution as determined by flow cytometry.

### 2.14. Single-Cell RNA Sequencing (scRNAseq) Cluster Identifications and Gene Analyses

The raw sequencing data for the Chromium 10× Genomics libraries were processed using Cellranger software v8.0 [43]. Then, the filtered gene-barcode matrices which contain barcodes with the Unique Molecular Identifier (UMI) counts that passed the cell detection algorithm were used for further analysis. The downstream analyses were performed using the Seurat single-cell data analysis R package [44]. First, the cells were demultiplexed with hashtag oligos (HTOs) and assigned to the corresponding sample. Then, the normalized and scaled UMI counts were calculated using the SCTransform method. We performed principal component analysis (PCA) and tSNE for dimensionality reduction using the highly variable genes. Data clustering was identified using the shared nearest neighbor (SNN)-based clustering on the first 30 principal components. Additionally, we assigned cell cycle scores for S and G2/M phases based on the well-defined gene sets to check if the clustering results were correlated with cell cycles. A single R package was utilized to identify the cell types using ImmGen mouse reference datasets. The marker genes were identified for each group of cells of interest. The heatmaps of selected genes for the cell types of interest were generated using pheatmap from the R package 1.0.12.

### 2.15. Single-Cell RNA Sequencing (scRNAseq)

CD11b^+^ magnetic bead-isolated cells from the spleens and livers of mice bearing 4T1 liver metastases (2 mice pooled) 5 days after treatment with either the vehicle or CBL0137 were washed once in DPBS with 0.01% BSA, and then stained with TotalSeq™-A0301 or 310 anti-mouse hashtag antibody at a 1:50 dilution, using antibody-clone barcode sequences such as A0301-M1/42-ACCCACCAGTAAGAC, A0305-M1/42-CTTTGTCTTTGTGAG, and A0310-M1/42-CCGATTGTAACAGAC (BioLegend). The cells were resuspended to a concentration of about 1000 cells/μL and loaded onto the 10× Genomics Chromium platform for droplet-enabled scRNAseq according to the manufacturer’s instructions. Single-cell libraries were generated using the 10× Genomics platform. Cells were loaded into the Chromium Controller (10× Genomics) where they were partitioned into nanoliter-scale gel beads-in-emulsion with a single barcode per cell. Reverse transcription was performed, and the resulting cDNA was amplified. Amplified cDNA was separated into full-length and feature barcode fractions using SPRISelect beads (Beckman Coulter, Pasadena, CA, USA). The full-length amplified cDNA was used to generate libraries by enzymatic fragmentation, end repair, a-tailing, adapter ligation, and PCR to add Illumina-compatible sequencing adapters. Feature barcode-derived cDNA was PCR-amplified to incorporate Illumina adapter sequences and unique sample indexes. The resulting libraries were evaluated on a D1000 screentape using TapeStation 4200 (Agilent Technologies, Santa Clara, CA, USA) and quantitated using a Kapa Biosystems qPCR quantitation kit for Illumina. They were pooled, denatured, and diluted to 300 pM with 1% PhiX control library added. The pooled library was then loaded into the appropriate NovaSeq Reagent cartridge and sequenced on NovaSeq6000 following the manufacturer’s recommended protocol (Illumina Inc., San Diego, CA, USA).

### 2.16. Data Availability

RNA-seq data were deposited in the National Center for Biotechnology Information’s Gene Expression Omnibus under accession no. GSE225504.

### 2.17. ELISA for IFN-γ

Mouse plasma from the blood and hepatic leukocytes in the RIP A buffer (Sigma-Aldrich, St. Louis, MO, USA) were stored at −80 °C for 2 weeks. Levels of IFN-γ were measured using an ELISA kit (R & D systems, Minneapolis, MN, USA).

### 2.18. Statistical Analysis

Comparisons between groups were performed using Student’s *t*-tests, and significance was reported at * *p* < 0.05, ** *p <* 0.01, and *** *p <* 0.001. A log-rank (Mantel–Cox) test was used for survival comparisons between groups using GraphPad Prism, Version 7 software (GraphPad Software, Inc., La Jolla, CA, USA).

### 2.19. Study Approval

All experimental protocols were approved by the Roswell Park Comprehensive Cancer Center Animal Care and Use Committee.

## 3. Results

### 3.1. CBL0137 Rapidly Decreases Tumor Cell Viability In Vitro

We took a reductionist approach to test the anti-tumor effects of single treatments of CBL0137 versus 5-FU in CT26 colorectal and 4T1 breast carcinoma murine tumor lines that have similar aggressive growth in vitro and in vivo and represent common liver metastases clinically. However, these murine tumors differ in their TME immune contexture, as CT26 responds to immune checkpoint inhibitors while 4T1 is considered cold and mimics triple-negative breast cancer which is highly resistant to immune checkpoint therapies [45,46]. One of the primary causes of refractory responses to immunotherapy is the robust expansion of MDSCs [47]. Thus, the chosen tumor lines also allow for comparisons beyond tumor-intrinsic drug effects in in vivo systems. To investigate the impact of CBL0137 on viability, MTT assays were performed on CT26 and 4T1 cells in vitro. After a single 15 min exposure to CBL0137 that was then washed away, tumor cells showed a significant reduction 3 h later in cell viability compared to untreated controls or 5-FU-treated groups (Figure 3A). The effect of CBL0137 was rapid, suggesting a fundamental difference to standard chemotherapies like 5-FU that induce mitotic catastrophe and are dependent upon the cell cycle. To assess the previously defined molecular functions of CBL0137, insoluble pellets containing chromatin-bound proteins and soluble nuclear proteins were evaluated by Western immunoblotting for FACT redistribution due to C-trapping [33,35] in CT26 and 4T1 cells. In vitro treatment of tumor cells with CBL0137 resulted in FACT binding to chromatin as evidenced by SSRP1 loss from the soluble fraction and accumulation within the chromatin pellet in treated samples (Figure 3B). Quantification of C-trapping demonstrated significant differences compared to controls and was highly sensitive with marked C-trapping at low doses of tested CBL0137. Confirming efficient drug uptake in tumor cells with immediate DNA interaction and induction of conformational changes, increased Z-DNA formation as a marker for CBL0137 DNA binding [33] was detected in the nucleus of both CT26 and 4T1 cells treated for 15 min with CBL0137 in vitro (Figure 3C). Together, these results indicate a rapid and effective reduction in tumor cell viability by previously described mechanisms of action of CBL0137 and a rationale for exploration in murine models of HAI in vivo.

### 3.2. CBL0137 HAI Induces Profound DNA Changes in Tumor Cells In Vivo

Mice were inoculated with CT26 or 4T1 tumor cells via a hemisplenectomy procedure to establish a model for diffuse liver metastases [37]. HAI consisting of the control (D5W), 5FU, or CBL0137 was administered for 15 min on day 10 post-tumor implantation. Similar to in vitro results, C-trapping occurred in both CT26 and 4T1 tumors following HAI with CBL0137, as noted by a substantial redistribution of SSRP-1 from soluble to pellet fractions in treated tumors (Figure 4A). As anticipated, the normal liver did not have any detectable SSRP-1, consistent with the observation of FACT overexpression in tumor cells [48] (Figure 4A). CBL0137 HAI-treated groups had significantly higher Z-DNA than control or 5-FU-treated groups, with 75% of CT26 and 60% of 4T1 tumor cells exhibiting Z-DNA formation following a 15 min exposure to CBL0137 (Figure 4B). To measure the downstream consequences of CBL0137 exposure in vivo during HAI, the ability to inhibit NF-κB signal transduction was measured using CT26 NF-kB-Luc or 4T1 NF-kB-Luc cells established in the liver via a hemisplenectomy technique. Luciferase was generated in mice by inducing NF-kB signaling with LPS, and the baseline of NF-kB activity was measured 30 min later. Significant inhibition of NF-κB activity was detected in CT26 and 4T1 tumors after CBL0137 HAI compared to D5W controls (Figure 4C). These results show that HAI administration of CBL0137 was capable of generating C-trapping and fundamental conformational changes in tumor cell DNA and inhibiting NF-kB activity within established liver tumors.

### 3.3. CBL0137 HAI Has Minimal Toxicity and Produces Clinical Anti-Tumor Responses in 4T1 Liver Cancer

Clinical outcome experiments were performed to determine toxicity levels and the degree of anti-tumor response associated with HAI. Clinically, drug delivery via HAI can be associated with direct toxicity to the liver parenchyma, pancreas, or distal stomach from inadvertent chemotherapy exposure [49]. CBL0137 was associated with a significant elevation of peripheral blood markers of liver toxicity AST and ALT 24 h following infusion as compared to 5-FU infusion but resolved by 48h with the mean levels below established toxicity thresholds (Appendix A). Measurements of amylase as a surrogate for pancreatic toxicity showed no consistent amylase elevation associated with HAI using CBL0137 vs. 5-FU (Appendix A). Given the early rise in amylase in D5W control groups, there was likely non-specific pancreatitis associated with the procedure that was transient and previously reported for this model system [38]. Lastly, as a global marker of toxicity, mouse body weight was followed after HAI. After an initial drop in weight immediately following the procedure across all groups, the weights of mice in all groups normalized by 2 weeks after the procedure (Appendix A). Collectively, these results indicate minimal overall toxicities associated with CBL0137 HAI.

The anti-tumor response was examined in metastatic models of CT26 luciferase and 4T1 luciferase reporter cell lines treated with either the control (D5W), 5-FU, or CBL0137 HAI. Using luminescent signal as a surrogate for tumor burden, the CT26 model had a trend toward smaller tumor volumes in 5-FU- and CBL0137-treated groups compared to controls, but no statistical differences were noted at any time point following HAI (Figure 5A). As a further measure of clinical response, quantification of TUNEL-positive tumor cells was determined by histological examination of livers 5 days after treatment. CT26 tumors trended toward increased TUNEL-positive cells in CBL0137-treated mice, but significant differences were not detected between treatment groups and untreated controls (Figure 5B). Survival curves were compared across the control and 5-FU- and CBL0137 HAI-treated groups. Consistent with the tumor volume and TUNEL observations, no survival benefit was noted across any group compared to controls in the CT26 system (Figure 5C). Contrary to these findings, significant tumor growth inhibition was noted by day 5 after HAI in 4T1 tumors receiving CBL0137 as compared to the control group (Figure 5A). Paralleling these tumor volume results in 4T1 liver tumors, CBL0137 induced significantly higher levels of apoptosis than 5-FU or control groups (Figure 5B). The CBL0137 HAI approach was necessary to produce this pattern of 4T1 tumor cell apoptosis in the liver as systemic delivery of CBL0137 via the tail vein was not sufficient (Appendix A). In the 4T1 system, a modest but significant survival benefit was noted only in CBL0137 HAI-treated mice (Figure 5C).

The prominent in vitro effects induced by CBL0137 on CT26 and 4T1 cell lines are in stark contrast to the lack of in vivo survival benefits in CT26 and only moderate survival benefits seen in 4T1. Since CBL0137 reaches tumor cell targets and induces significant DNA changes in vivo, we hypothesized that the host immune system may have influenced the observed clinical outcomes. Therefore, the dominant immune phenotypes inherent to CT26 and 4T1 tumors were exploited to dissect potential indirect factors.

### 3.4. CBL0137 HAI Profoundly Generates a Favorable MDSC/T Cell Ratio

In terms of immune contexture, CT26 tumors are low in MDSCs and are considered immune-responsive while 4T1 tumors have high levels of MDSCs and are resistant to checkpoint inhibitor immunotherapy [50,51,52]. One of the key mechanisms of 4T1 tumors that makes them refractory to immunotherapy is the robust expansion of MDSCs and the subsequent inhibition of CD8^+^ T cell infiltration [53]. We define MDSCs by a CD11b^+^Gr1^+^ phenotype and the ability to suppress T cell proliferation as previously reported [54], with about 80% of the enriched CD11b^+^ cells co-expressing Gr-1^+^ with effective T cell suppression (Appendix A). Immunogenic CT26 tumors are noted to have minimal MDSC expansion and increased baseline CD8^+^ T cell infiltration which was evident following control (D5W) HAI (Figure 6A). 5FU HAI did not demonstrate any impact on MDSCs or T cell infiltration in CT26 tumors as compared to the control HAI (Figure 6B,C). CBL0137 HAI did not influence the already low levels of MDSCs found in CT26 tumors (Figure 6B) but did slightly increase the level of CD8^+^ T cell infiltration in CT26 tumors (Figure 6C), and this was reflected in a significant decrease in MDSC/CD8^+^ T cell ratios [ratio of 4.5:1 (vehicle) to 0.8:1 (CBL0137)] (Figure 6D). In contrast, liver 4T1 tumors at baseline following control (D5W) HAI have extensive intratumor MDSCs with exclusion of CD8^+^ T cells (Figure 6A). While 5FU HAI had no appreciable changes in MDSCs or T cells in 4T1 tumors, CBL0137 HAI significantly reduced intratumoral MDSCs (Figure 6A,B) and produced a significant increase in CD8^+^ T cell infiltration in previously cold 4T1 tumors (Figure 6C). The profound increase in tumor-infiltrating CD8^+^ T cells could be at least partially attributed to a simultaneous reduction in MDSCs in 4T1 tumors treated with CBL0137 HAI as reflected in the favorable MDSC/CD8^+^ T cell ratios [ratio of 14.7:1 (vehicle) to 2.2:1 (CBL0137)] (Figure 6D).

The influence of CBL0137 HAI on immune cell subsets was not limited to MDSCs and CD8^+^ T cells but also skewed CD4^+^ T cell compartments towards a favorable anti-tumor phenotype. Specifically, CBL0137 HAI significantly increased the levels of IFN-γ-positive Th1 CD4^+^ T cells in treated livers compared to controls (D5W) in both CT26 and 4T1 tumors (Appendix A). Interestingly, CD4^+^ Treg cells did not significantly increase concomitantly with CBL0137 HAI, but 5-FU HAI trended toward a Treg increase in CT26 tumors and significantly increased Tregs in 4T1 tumors (Appendix A). The kinetics of CBL0137 HAI-mediated immune cell changes in the 4T1 model showed a reduction in MDSCs and a concurrent increase in T cell populations peaking at 3 days post-infusion with a return to baseline levels by day 7 (Appendix A). Compared to conventional 5-FU chemotherapy, these results collectively suggest that CBL0137 HAI may influence outcomes beyond direct anti-tumor responses by inducing a transient, but significant, decrease in MDSCs. We also evaluated the effectiveness of HAI compared to intravenous systemic treatment using CBL0137 in 4T1 liver metastases. HAI led to a significantly greater reduction in MDSCs and lower MDSC/CD8^+^ T cell ratios, whereas the systemic treatment group showed efficacy only in the blood, with limited impact on the liver (Appendix A).

### 3.5. CBL0137 HAI Acts Synergistically with Anti-PD-1 Immunotherapy

To stringently examine the potential for synergy with HAI, anti-PD-1 Ab was administered over multiple days as reported previously for therapeutic efficacy in the CT26 model [55]. Even though anti-PD-1 therapy as a single agent showed a strong trend toward delaying tumor growth regardless of the anatomic site of the tumor (Appendix A), the combination of anti-PD-1 with either CBL0137 HAI or 5FU HAI did not improve survival compared to anti-PD-1 alone in the CT26 model (Figure 7A). A closer inspection of the infiltrating CD8^+^ T cell subsets following HAI and anti-PD-1 combined treatment did not demonstrate any significant changes in T cell phenotypes in the CT26 models across the various groups (Figure 7B). The only changes noted in T cell subsets in the CT26 model were in mice treated with 5-FU HAI alone which was associated with a reduced number of CD8^+^ and CD4^+^ effector memory T cells and likely represented chemotherapy depletion (Appendix A).

In the 4T1 model, the profile of immune changes in T cells recovered from the liver after CBL0137 HAI is marked by a high expression of PD-1 and IFN-γ (Appendix A). Given these favorable immune features in addition to the previously demonstrated MDSC depletion, the potential for CBL0137 HAI synergy with anti-PD-1 immunotherapy was similarly explored in the 4T1 model which is refractory to PD-1-/PD-L1-targeted immunotherapy [52]. Consistent with 4T1 resistance to immunotherapy, no mice survived beyond 25 days when treated with anti-PD-1 alone (Figure 7A) which was identical to the survival of vehicle control groups noted previously (Figure 5C). Moreover, the addition of anti-PD-1 to standard 5-FU-based HAI had no discernable impact on survival compared to anti-PD-1 alone in the 4T1 model (Figure 7A). However, the addition of CBL0137 HAI to anti-PD-1 treatment partially overcomes resistance in the 4T1 tumor model with significantly improved survival (Figure 7A). CBL0137 HAI and anti-PD-1 combined treatment demonstrated a significant increase in CD8^+^CD44^+^CD62L^−^ effector memory cells in 4T1 tumors with an overwhelming majority of CD8^+^ T cells and CD4^+^ T cells representing effector memory subtypes (Figure 7B). Suggesting true synergy, anti-PD-1 treatment alone did not skew the baseline T cell subsets (Figure 7B) and CBL0137 HAI only influenced the CD8^+^ T cell population toward an effector memory subtype (Appendix A). The addition of anti-PD-1 to 5-FU HAI had no impact on T cell subset phenotype (Figure 7B), nor did 5-FU HAI alone (Appendix A) in the 4T1 model.

### 3.6. CBL0137 HAI Targets a Specific Subclass of MDSCs via Z-DNA Induction

Single-cell RNA sequencing (scRNAseq) was used to further characterize the MDSC changes noted during CBL0137 HAI treatment in 4T1 tumors. Consistent with our previous data and compared to control groups, depletion was not noted within T cells, B cells, NK cells, or DC populations (Figure 8A), but the elimination of a large MDSC population was noted after CBL0137 HAI (Figure 8B). Focusing on this population by cell sorting for CD11b^+^ cells and performing scRNAseq, the loss of a specific subset of MDSCs was noted that expressed IL1b and CD84, which are associated with inflammation and adhesive interactions with T cells, respectively (Figure 8C). Specifically, MDSCs expressed significantly lower levels of IL1b (*p* < 0.0001) and CD84 (*p* < 0.01) transcripts following CBL0137 injection compared to the vehicle-injected group (Figure 8D). To examine the mechanism of selective depletion of MDSCs associated with CBL0137 HAI, cell sorting of MDSCs from the liver or spleen of 4T1 tumors grown subcutaneously or in the liver underwent Western blot analyses of Z-DNA-sensing molecules. The consistent and high expression of the Z-DNA-binding protein 1 (ZBP1) and its mediator of necroptotic cell death, receptor-interacting protein kinase 3 (RIPK3), in MDSCs suggested a direct mechanism for selective depletion from Z-DNA induction by CBL0137. The molecules necessary for MDSC necroptosis were present whether the MDSCs were recovered from the liver or spleen and were not dependent upon whether the tumor was implanted in the subcutaneous or liver site (Figure 8E and Appendix A). Unlike the rapid induction of Z-DNA in tumor cells within 15 min of exposure to CBL0137 (Figure 3C), in vitro induction of Z-DNA was initially noted at 30 min (Appendix A) in MDSCs while peaking at 60 min, and importantly, it was nearly absent in T cell populations (Figure 8F and Appendix A). T cells recovered from mice with 4T1 tumors have the same necroptosis machinery as MDSCs (Appendix A) but were resistant to Z-DNA formation by CBL0137 (Figure 8F and Appendix A).

In addition, the detailed quantification of different cell populations after vehicle or CBL0137 HAI was analyzed, and depletion of PMD-/M-MDSC populations was detected with increased macrophages and T, B, and NK cells in the liver (Figure 9A). We also analyzed specific gene expression in existing MDSCs after HAI, and significantly high levels of Camp, Ltf, and Ly6G expressions representing a mature neutrophil state were found in the CBL0137-treated group compared to the vehicle group; gene expressions involved in immune suppression such as Wfdc17, S100a8/9, Il1b, and Cd84 were decreased significantly in the CBL0137 group. And gene expressions of Cebpe and Tuba1b for granulocytic differentiation, indicating the existence of a proliferative pool of neutrophils, were increased with CBL0137 treatment (Figure 9B). We have shown that CBL0137 suppressed NF-κB transcription within various cancer cells resulting in p53 activation [26]. NF-κB-related genes and p53 promoter suppressor genes such as Csf1, Fxyd5, and Jun were suppressed in MDSCs after CBL0137 HAI (Figure 9B). However, we could not detect an increase in IκBα (Nfkbia) gene expression in the CBL0137-treated group compared to the vehicle in MDSCs. Furthermore, multiple genes that were responsible for homeostasis, proliferation, and activation in T cells were highly expressed after CBL0137 HAI treatment, including Eif6, Traf2, Tab2, Ckb, Baz1b, Lcp1, Ago2, Rap1a, Icam2, and Dok2. Th1 transcription genes such as Gadd45b, Itgb1, Ifi27l2a, and Tbx21 were especially overexpressed in the CBL0137 HAI-treated group. Simultaneously, genes promoting anti-cancer efficacy such as Cd160, Cd244a, and Ccr2 were highly expressed while a relay gene of the NF-κB pathway, Card11, was suppressed in NK cells after CBL0137 treatment. Lastly, Socs3 is a negative regulator of NK activity and therapeutic targeting of Socs3 in NK cells potentiates the killing of tumor targets [56]. CBL0137 suppressed this Socs3 expression in NK cells (Figure 9B).

Collectively, these data demonstrate the ability of CBL0137 HAI to selectively deplete MDSC populations that inhibit anti-tumor immune responses, simultaneously maintain T cell populations, and overcome anti-PD-1 resistance to improve survival in the immunologic cold 4T1 liver metastatic model.

## 4. Discussion

The management of metastatic liver disease continues to evolve with aggressive measures leading to improved clinical outcomes. Unfortunately, the majority of metastatic liver patients are not amenable to curative surgical extirpation, and systemic therapy options or ablative techniques do not typically result in a cure. The recent introduction of an immune checkpoint blockade has revolutionized the treatment of various cancers and has highlighted the role of the immune system in oncogenesis and cancer therapy. While promising, the vast majority of patients with liver metastases do not respond to standard immunotherapy options [57,58]. Novel therapies to render cold tumors immunotherapy-responsive are needed for this large population of patients. HAI has demonstrated clinical efficacy in select patients, but reliance on standard chemotherapy agents that concurrently deplete immune cells limits the full potential of this therapy and impedes the expansion of anti-tumor immune responses.

Similar to our prior work in the intra-arterial delivery of CBL0137 in other tumor models [59], the current in vitro and in vivo HAI studies showed that CBL0137 was rapidly taken up by CT26 and 4T1 tumor cells and caused an immediate decrease in cell proliferation. Z-DNA is one possible double helical configuration of DNA and is a left-handed structure in which the helix winds to the left in a zigzag pattern [60,61]. Treatment of tumor cells with CBL0137 caused the appearance of nuclear Z-DNA in a dose- and time-dependent manner in vitro [33]. For in vivo experiments, Z-DNA was detected 2 h after a 15 min CBL0137 HAI in CT26 and 4T1 tumors, and after chromatin trapping as noted by the appearance of SSRP1, a subunit of FACT complex, in the insoluble chromatin fraction within 1 h as previously described [33]. These findings suggested that CBL0137 absorption in the target tissue, metastatic liver lesions, was rapid. Also, as a direct result of C-trapping by CBL0137, there was subsequent inhibition of NF-κB which is an important transcription factor associated with the expression of pro-inflammatory cytokines and the upregulation of tumor survival genes [26]. CBL0137 acted directly on cancer cells in a non-genotoxic manner by trapping the FACT complex within DNA complexes, leading to a reduction in NF-κB activity, and resulting in cell apoptosis and necroptosis as demonstrated in these data and prior work [26,62]. The quick uptake, liver tumor-specific affinity, and sparing of normal liver parenchyma are essential characteristics of the ideal drug for use in a focused, liver-directed regional therapy for liver metastases. Even more importantly, CBL0137 HAI was well tolerated, making it superior to standard HAI chemotherapies that are commonly related to sclerosing cholangitis or chemical hepatitis [49], which is relevant to patients with heavy disease burden and limited functional liver reserve.

We postulate that suppressing MDSCs in the liver will result in better survival outcomes in patients with liver metastases treated using immune checkpoint blockade therapy. However, the problem stems from the inability to deliver efficient drug concentrations to the liver without delivering harmful doses to sensitive normal tissues [63]. Systemic anti-MDSC treatments have limited ability to decrease MDSCs in the liver. Moreover, systemic treatments with conventional chemo agents for the elimination of MDSCs can not only have a direct cytotoxic effect on tumor cells but can also eliminate T cells which are critical for immunotherapy. As an alternative to systemic therapy, directed local delivery of a drug to the liver via a regional artery, HAI, with a fractional dose of systemic drug usage could be a solution to reduce MDSC burden while sparing T cells in the liver. HAI takes advantage of the access to liver lesions provided by the hepatic arterial system and the first-pass effect [64].

Greater apoptosis, delayed tumor growth, decreased tumor volumes, and better survival outcomes were detected with CBL0137 HAI treatment compared to the vehicle or 5FU in mice bearing 4T1 breast cancer liver metastasis, but this anti-tumor effect was not as pronounced in mice bearing CT26 colon cancer liver metastasis. Despite evidence of CBL0137 having direct effects on CT26 and 4T1 tumor deposits in the liver during HAI, the survival benefit was either absent or modest, respectively. Due to this observed discordance, we explored the possibility that different pathophysiologies, specifically previously described immune suppressor networks, may influence outcomes between the CT26 colon and 4T1 breast cancers that were examined. Confirming these differences and in line with prior reports [65,66,67], 4T1 liver tumors were highly enriched for MDSC populations, as compared to CT26 liver tumors. MDSCs are capable of supporting tumor growth through remodeling of the tumor microenvironment [68]. While MDSCs play a crucial role in tumor survival in the 4T1 model, they are less critical for CT26 tumor survival. As a result, the elimination of MDSCs using 5-FU or CBL0137 in the CT26 model showed limited immune efficacy.

A growing body of evidence suggests that tumor infiltration of immune-suppressive MDSCs correlates with poor prognosis and drug resistance [69,70,71,72]. We demonstrated that the treatment of 4T1 liver metastasis with CBL0137 HAI resulted in the selective depletion of MDSCs in liver tumors along with a significant increase in the levels of Th1 cells. CBL0137 HAI had the advantage of not inducing higher levels of Tregs as was the case after 5-FU HAI treatment and as was reported previously for systemic 5-FU treatment in 4T1 models [73]. The observed reduction in immune suppressive MDSCs and the increase in the Th1 cell population were associated with an increase in CD8^+^ T cell infiltration into the tumor microenvironment after CBL0137 HAI. These findings are clinically relevant as increased CD8^+^ T cell infiltration has been directly correlated with decreased recurrence and improved survival across several tumor types including breast cancer [74]. We leveraged these findings by showing that the combination of CBL0137 HAI followed by anti-PD-1 mAb expanded CD4 and CD8 effector memory populations and increased survival in the immune refractory 4T1 model.

CBL0137 selectively depleted MDSC subpopulations expressing *Il1b* and *CD84* but did not affect other immune cells including T, B, and NK cells. *Il1b* and *CD84* are important shared genes involved in immune suppressive features such as a T cell-suppressive capacity and increased reactive oxygen species (ROS) production between PMN- and M-MDSCs [75,76,77,78,79]. Our data indicate that the selective MDSC depletion is mediated via CBL0137 induction of Z-DNA and the downstream consequences of Z-DNA detection (ZBP1) combined with known mediators of necroptosis (RIPK3) as previously described for viral infections [62] and as most recently described for tumor-associated fibroblasts [80]. Z-DNA formation requires all necessary machinery such as ZBP-1, RIPK3, and MLKL to induce necroptosis in MDSCs. Although Z-DNA formation in MDSCs is not extensively studied, Z-DNA forms under conditions of negative supercoiling, high transcriptional activity, and specific sequence contexts [80,81,82,83]. MDSCs exhibit unique transcriptional profiles that support their role in immune suppression [75]. Z-DNA may form in areas where genes related to metabolic reprogramming, ROS production, and immunosuppressive cytokine release are highly transcribed. And our data demonstrated that the *Il1b^+^CD84^+^* subpopulation of MDSCs, which have enhanced suppressive capabilities in immune suppression compared to *Il1b^−^CD84* MDSCs, were selectively depleted by CBL0137. Importantly, T cells are concurrently maintained and do not appear to be susceptible to Z-DNA formation by CBL0137, despite having the necessary machinery to execute necroptosis. Due to their critical role in host defense, it is postulated that T cell DNA might be resistant to Z-DNA formation and subsequent necroptosis, similar in principle to a recently discovered mechanism of germline genome safeguarding specifically against Z-DNA formation [84]. The time course in vitro and in vivo for the induction of Z-DNA is more rapid in tumor cells compared to MDSCs and while a single dose of CBL0137 in vivo is capable of bulk depletion of MDSCs, this effect was transient peaking at 3 days post-infusion with a rebound of MDSCs to baseline levels by day 7. The mechanisms defined for MDSC depletion may represent a relevant clinical biomarker for CBL0137 response in MDSC-related tumors, and the kinetics suggest that the addition of immunotherapy would be optimal at the maximal time of depletion.

Our study reflects only a single HAI treatment along with a PD-1 blockade as a reductionist approach for a stringent investigation of interactions. And various genes related to proliferation and activation in T and NK cells were overexpressed after CBL0137 treatment. These cellular effects of CBL0137 suggest a novel mechanism of augmentation of PD1 blockade efficacy as well. Thus, further evaluation will be needed to determine if repeated or continuous treatment of CBL0137 will confer additional therapeutic effects. Clinically, HAI pumps or multiple hepatic artery percutaneous infusions can deliver continuous or multiple doses of CBL0137 and may further magnify the observed anti-tumor results. Importantly, the toxicity of a single infusion in our study appears tolerable, but a continuous or repeated infusion may have cumulative effects that cannot be determined from these investigations. Similarly, only one dose of anti-PD-1 mAb was used with the same injection schedule which may underestimate the effects of different schedules and doses.

In summary, CBL0137 HAI could be a viable option for the clinical treatment of liver metastases and serve as a foundation for future combinations or sequential therapies that take advantage of this unique immune potentiation.

## 5. Conclusions

CBL0137 delivered via HAI was tumoricidal to liver metastasis with minimal morbidity and highly favorable anti-tumor immune properties not seen with standard chemotherapies. CBL0137 HAI may not only directly treat liver metastases, but also “precondition” tumors to render them sensitive to current immunotherapies.

## Figures and Tables

**Figure 1 cancers-16-03711-f001:**
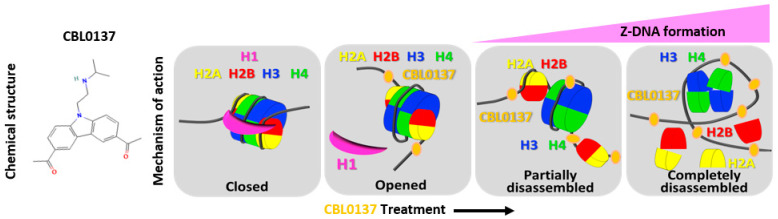
A schematic diagram illustrating the mechanism of action involving chromatin decondensation and the chemical structure of CBL0137. Z-DNA formation occurs following the interaction of CBL0137 with chromatin, leading to histone disassembly.

**Figure 2 cancers-16-03711-f002:**
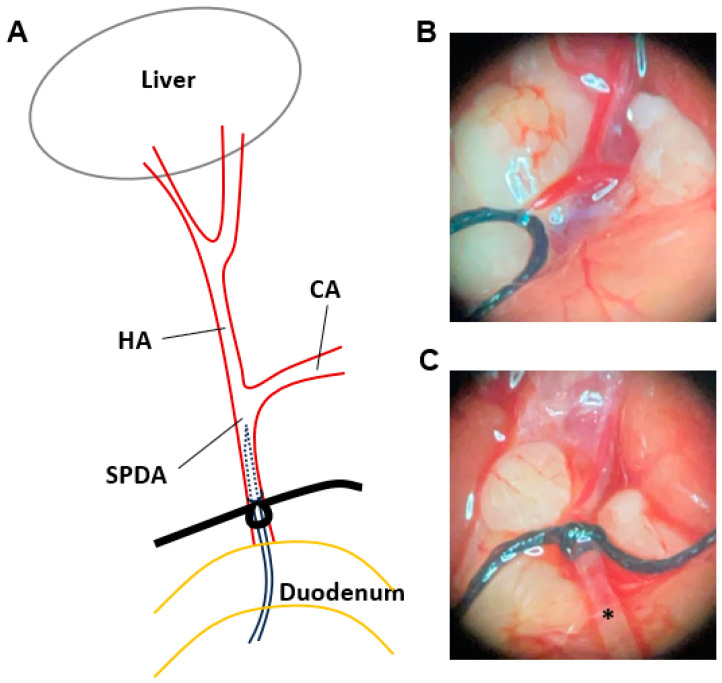
A schematic illustration and representative photographs of HAI via the superior pancreaticoduodenal artery (SPDA). (**A**) A diagram of the HAI procedure via the SPDA. (**B**) A representative photograph of the vascular anatomy before cannulation. (**C**) A representative photograph after catheter placement (*) in the SPDA.

**Figure 3 cancers-16-03711-f003:**
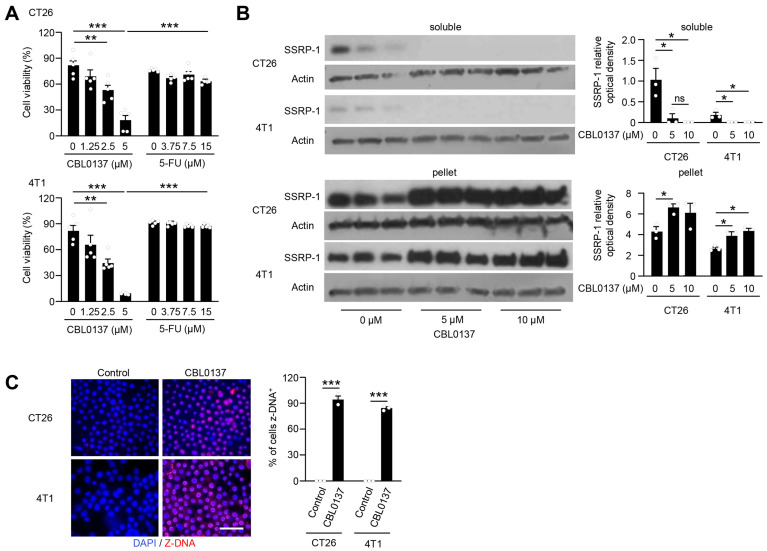
CBL0137 rapidly limits proliferation of CT26 and 4T1 tumor cells in vitro by altering DNA/histone interactions. (**A**) In vitro cell viability was significantly reduced by a 15 min treatment with CBL0137 as compared to 5FU (*n* = 5). Representative data among 3 independent experiments. **: *p* < 0.01; ***: *p* < 0.001 from Student’s *t*-test; error bar: standard error of the mean. (**B**) Western blot analysis of SSRP 1 redistribution after in vitro treatment with different doses of CVB0137. As a marker for CBL0137 activity, SSRP1 loss in the soluble fraction of nuclear proteins and accumulation in the pelleted fraction of proteins from tumor cells was noted in vitro. Representative data between 2 independent experiments. Results were quantified using 1.53v ImageJ. *n* = 3; *: *p* < 0.05 from Student’s *t-*test; error bar: standard error of the mean, ns: not significant. (**C**) Immunofluorescence (IF) of Z-DNA in cancer cells. CT26 and 4T1 cells were treated with either the control (D5W) or CBL0137 for 15 min in vitro and then were stained with Z-DNA antibodies showing marked induction of Z-DNA formation. Representative IF figures among 3 independent experiments. *n* = 3; bar is 50 µm; ***: *p* < 0.001 from Student’s *t*-test; error bar: standard error of the mean.

**Figure 4 cancers-16-03711-f004:**
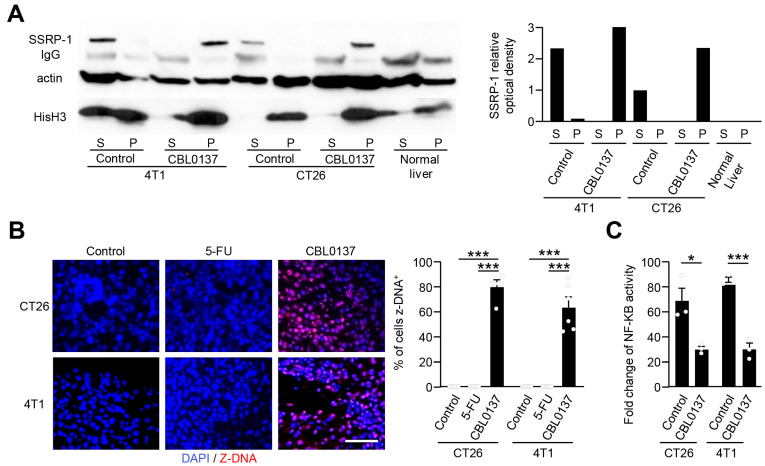
HAI with CBL0137 induces distinct chromatin decondensation with Z-DNA formation in vivo. (**A**) Western blot analysis of SSRP 1 redistribution after in vivo treatment via CBL0137 HAI. HAI with CBL0137 induced DNA/histone changes as noted by loss of SSRP-1 in soluble fractions and appearance in the pellets of 4T1 and CT26 liver tumors. Single experiment. (**B**) Z-DNA detection using IF after in vivo HAI. In vivo HAI with CBL0137 induced nuclear changes as depicted by the generation of Z-DNA as compared to control (D5W) or 5-FU. Representative IF figures among 3 independent experiments. *n =* 6; ***: *p* < 0.001 from Student’s *t*-test; error bar: standard error of the mean. (**C**) Suppression of NF-κB expression 4 h after CBL0137 HAI compared to the control HAI group. Representative data from 2 independent experiments. *n =* 3, *; *p* < 0.05, ***; *p* < 0.001 from Student’s *t*-test, error bar; standard error of the mean.

**Figure 5 cancers-16-03711-f005:**
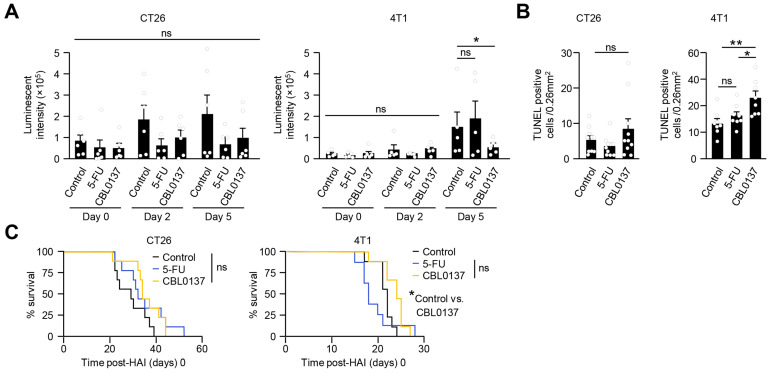
CBL0137 HAI produces different antitumor responses in CT26 and 4T1 liver metastases. (**A**) Luminescence 5 days after HAI with CBL0137 was reduced significantly in 4T1 liver tumors, but not CT26 liver tumors; representative data among 3 independent experiments, CT26 (*n =* 6) and 4T1 (*n =* 5). (**B**) Numbers of TUNEL-positive cells in tumor nodules. A trend to greater numbers of apoptotic cells and a statistical increase in apoptotic cells are identified by TUNEL stain 5 days after HAI treatment with CBL0137 in CT26 and 4T1 liver metastases. Pooled data with 3 independent experiments with CT26 tumor (*n =* 10) and 4T1 (*n =* 8) tumor. Each value represented an average of 3 slides per mouse. *: *p* < 0.05; **: *p* < 0.01 from Student’s *t*-test; error bar: standard error of the mean; ns: not significant. (**C**) Survival curves after HAI with different treatments. No survival benefit was detected in mice bearing CT26 liver metastasis. Alternatively, survival data showed statistically significant but modest survival benefits in mice bearing 4T1 liver metastasis. Pooled data from 3 independent experiments. *: *p* < 0.05; ns: not significant; *n =* 9, using log-rank tests.

**Figure 6 cancers-16-03711-f006:**
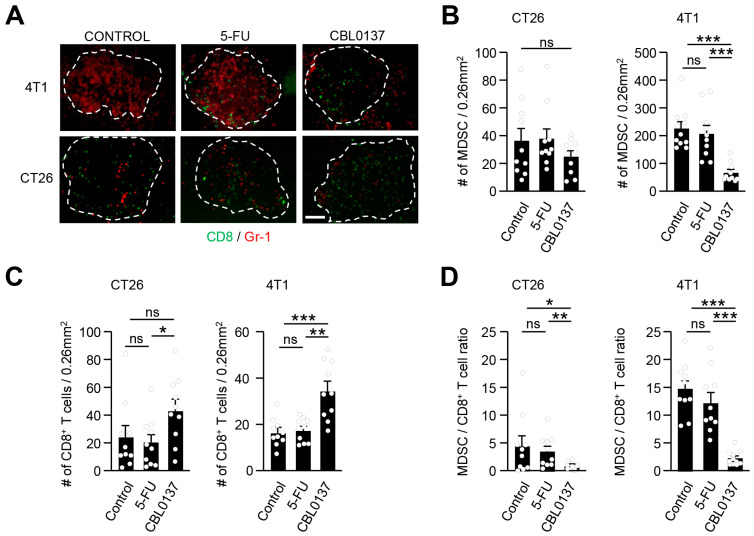
CBL0137 HAI exerts favorable changes in immune cell composition of metastatic tumors. (**A**) Representative IF images of CD8- and Gr-1-positive cells. CBL0137 reduced intratumoral MDSCs in both CT26 and 4T1 tumors. Gr-1^+^ (red) and CD8^+^ (green) antibodies 5 days after HAI with control (D5W), 5FU, or CBL0137. The scale bar is 50 μm; the white dashed line denotes the tumor boundary. (**B**) Numbers of MDSCs in tumor modules after HAI with different treatments in CT26 and 4T1 liver metastatic models. CBL0137 HAI decreased the number of MDSCs in 4T1 liver tumors but had no impact on the already low numbers of MDSCs in CT26 liver tumors. ***: *p* < 0.001 from Student’s *t*-test; error bar: standard error of the mean; ns: not significant. (**C**) Numbers of CD8^+^ T cells in tumor nodules after HAI. CD8^+^ cells tended to increase in CT26 tumor nodules and significantly increased in 4T1 tumor nodules from CBL0137 HAI. #; number, *: *p* < 0.05; **: *p* < 0.01; ***: *p* < 0.001 from Student’s *t*-test; error bar: standard error of the mean; ns: not significant. (**D**) The ratios of MDSCs and CD8^+^ T cells in tumor nodules. CBL0137 induces favorable statistically significant MDSC/CD8^+^ T cell ratios in CT26 and 4T1 tumors 5 days after HAI. Representative photos and pooled data from 3 independent experiments, with each value representing an average of 3 slides per mouse. *: *p* < 0.05; **: *p* < 0.01; ***: *p* < 0.001; *n =* 10, from Student’s *t*-test; error bar: standard error of the mean.

**Figure 7 cancers-16-03711-f007:**
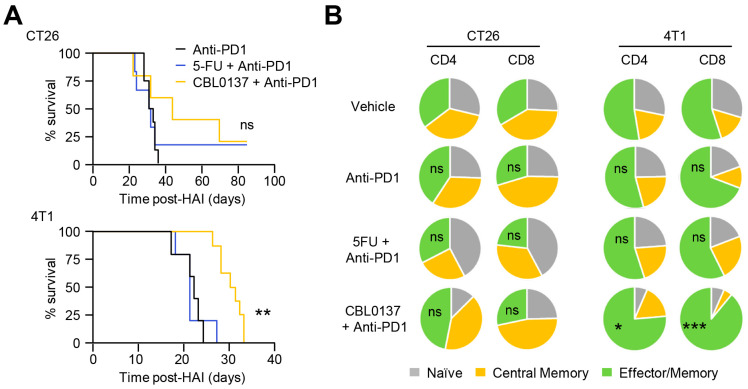
PD-1 blockade therapy augments survival of CBL0137 HAI. (**A**) Survival curves with anti-PD-1 therapy after HAI. Overall survival was improved, and cures generated by the addition of anti-PD-1 in HAI CBL0137 and 5-FU groups in CT26 tumor-bearing mice. And overall survival was significantly extended by the addition of anti-PD-1 therapy to CBL0137 HAI in 4T1 tumor-bearing mice compared to PD-1 alone. Survival data were acquired after each treatment with mice bearing 4T1 and CT26 liver metastases. Pooled data from 2 different experiments. **: *p* < 0.01; ns: not significant; *n* ≥ 5, using log-rank tests. (**B**) Compositions of CD4^+^/CD8^+^ naïve, central memory, and effector memory T cell populations. T cell-activated statuses were analyzed as naïve (CD62L^+^CD44^−^), central memory (CM; CD62L^+^CD44^+^), and effector memory (EM; CD62L^−^CD44^+^) cell populations 5 days after each treatment and showed significant skewing toward effector memory subsets in combined CBL0137 and anti-PD-1 treatment in 4T1 mice. Pooled data from 2 different experiments. *: *p* < 0.05; ***: *p* < 0.001; ns: not significant; *n* ≥ 4, from Student’s *t*-test.

**Figure 8 cancers-16-03711-f008:**
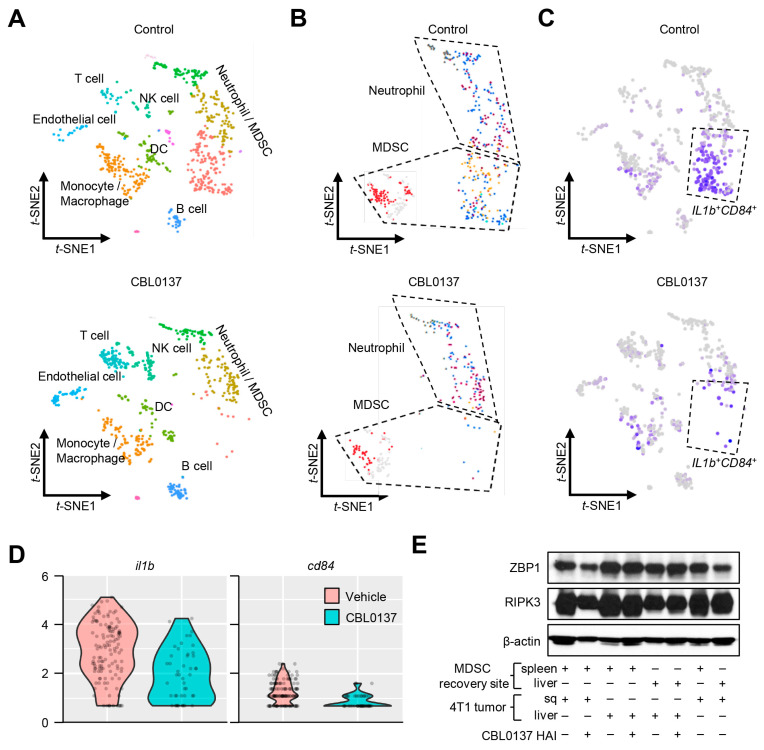
CBL0137 HAI selectively depletes *Il1b* and *CD84*-expressing hepatic MDSC populations. (**A**) ScRNAseq was performed with leukocytes from the liver 5 days after HAI with either vehicle or CBL0137. Cell clusters were characterized after vehicle or CBL0137 treatment. (**B**) Subcategorized neutrophil and monocyte populations based on ImmGen data. (**C**) Neutrophil and monocyte populations were identified with *Il1b* and CD84 genes after vehicle or CBL137 HAI. (**D**) Violin plots showing different levels of Il1b and Cd84 gene expression in existing MDSC populations after vehicle or CBL0137 treatment. (**E**) Western blot for the Z-DNA-sensing protein ZBP1 and a downstream molecule (RIPK3) in MDSCs recovered under different conditions (as depicted) shows consistently high levels of ZBP1 and RIPK3 in all MDSCs recovered. (**F**) IF images of Z-DNA in T cells and MDSCs. After 1 h of exposure to CBL0137 in vitro, Z-DNA is extensively detected in MDSCs but not T cells. Representative IF figures among 2 independent experiments. *n =* 3; **: *p* < 0.01 from Student’s *t*-test; error bar: standard error of the mean.

**Figure 9 cancers-16-03711-f009:**
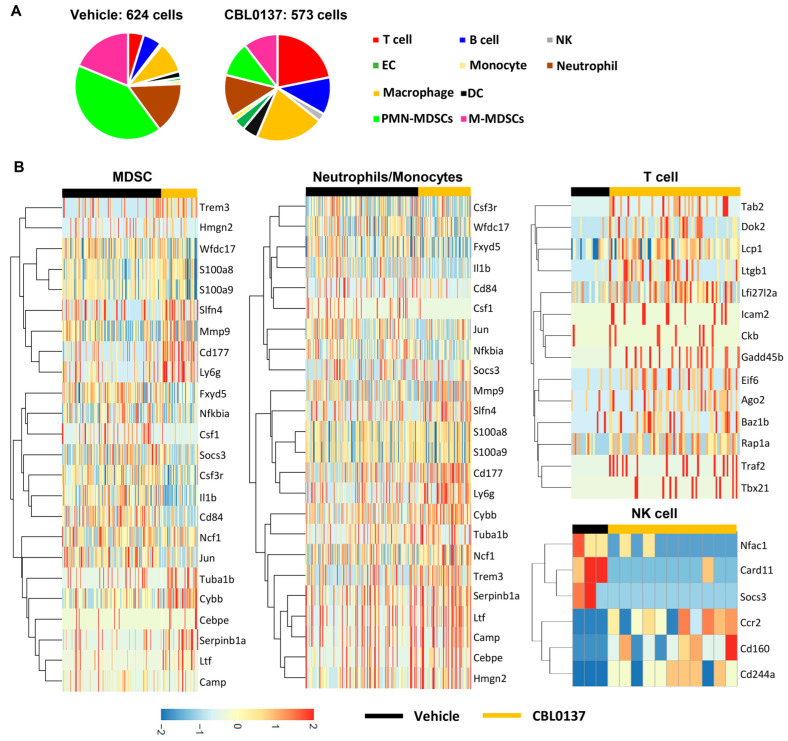
ScRNAseq data reveal differences in cell clusters and gene expressions. Isolated leukocytes from the liver 5 days after HAI with either vehicle or CBL0137 were analyzed to (**A**) characterize cell populations and (**B**) different gene expressions in MDSCs, neutrophils, monocytes, and T and NK cells.

## Data Availability

RNA-seq data were deposited in the National Center for Biotechnology Information’s Gene Expression Omnibus under accession no. GSE225504. The datasets used and/or analyzed during the current study are available from the corresponding author upon reasonable request.

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
