# Peer review of "Enhancing Anti-PD-1 Immunotherapy by Targeting MDSCs via Hepatic Arterial Infusion in Breast Cancer Liver Metastases"

_cancers, 2024, doi:10.3390/cancers16213711_

Round 1

Reviewer 1 Report

Comments and Suggestions for Authors

1.       line 109: The administration amount/schedule and clone ID of the anti-mouse PD-1 antibody must be described.

2.       Figure 3C: Is the difference between 5-FU and CBL0137 significant in mice bearing 4T1 cells?

3.       Figure 6E & Suppl. Figure 5A: These results indicate that necroptosis is NOT involved in the depletion of specific subset of MDSC by CBL0137 as MLKL remains completely unphosphorylated. If necroptosis is occurred, because Z-DNA formation is observed in MDSCs, why is phosphorylated MLKL not observed? Therefore, CBL0137 may depletes MDSCs by a different mechanism. Relevant sentences in results and discussion should be revised.

4.       Did the authors examine RIPK3 phosphorylation or PANoptosis?

5.       Necroptosis in cancer cells should be examined.

6.       Legends for supplementary figures should be provided.

Reviewer 2 Report

Comments and Suggestions for Authors

1.       The authors tried to show how MDSC depletion complemented the anti-tumor efficacy of CBL0137 HAI. This was further shown to be complemented upon combination with programmed cell death (PD)-1 blockade therapy. However, the title sounds as if the manuscript is focused at developing a novel method for MDSC depletion that aided in unlocking immunotherapy responses in cold hepatic metastases. The authors should revisit the title for a true reflection of the authors observations.

2.       Technically, there is no specific depletion strategy was used in the current manuscript. As reduction of MDSC number is a result of mode of action by CBL0137 on CT26, these statements need authors attention for clarity.

3.       Figure 2 was shown in the middle of the methods under T cell suppression assay. (Lines 204-214) and that too ahead of Figure 1.

4.       In lines 215-217, what the authors mean by “To evaluate T cell suppression during continuous exposure to MDSC, splenic and hepatic CD11b+ cells were purified from 4T1 liver metastasis-bearing BALB/c mice using anti-CD11b+ magnetic beads (Miltenyi Biotec) as the provided manual? Does it mean “as per the provided manual?”. There are several lines in the manuscript that needs authors attention for use of language.

5.       Images in Fig 1c, Fig 2B and those in Figure 6 and Suppl. Fig 5 are poor in clarity.

6.       In Figures 4B and 4C, the authors have shown that 5FU HAI did not demonstrate any impact on MDSC or T cell infiltration in CT26 tumors. However, PMID: 33604533 have shown MDSCs depletion using low-dose 5-fluorouracil (5-FU) chemotherapy. 

They also show that 5-FU mediated depletion of MDSC enhances T-cell infiltration and anti-tumor response in immunotherapy–resistant lung tumor. What are the authors take on the 5FU in CT26 tumors?

7.       The authors report that 4T1 tumors have high levels of MDSC compared to CT26 tumors, while CBL0137 acted on lowering the MDSC population and thus aiding in increased CD8 T cell infiltration. Therefore, it can be anticipated that the action of CBL0137 will be much strong in CT26 compared to high MDSC bearing 4T1 bearing tumors. However, it is not the case as the authors report that CBL0137 HAI did not influence the already low levels of MDSC found in CT26 tumors (Figure 4B) but did significantly increase the level of CD8+ T cell infiltration in CT26 tumors (Figure 4C).

Moreover, the activity of CBL0137 on CT26 is stronger in terms of cytotoxicity, in vitro. Do the authors think that antitumor activity of CBL0137 HAI in vivo is just by its actions on MDSC or could there be another possible mechanism of action?

8.       It was stated that the UMAP and tSNE were generated from the scRNAseq data, but no UMAP’s were shown in the figures!

Comments on the Quality of English Language

none

Reviewer 3 Report

Comments and Suggestions for Authors

Acute leukemia is the most malignant group of tumors of the hematopoietic system. This group includes acute lymphoblastic leukemia, originating from cells of the lymphoid lineage of hematopoiesis, and acute myeloid leukemia, originating from cells of the myeloid series. Over the past decades, significant advances have been made in the treatment of adolescent acute lymphoblastic leukemia, with 5-year relapse-free survival increasing to 70%. Nevertheless, acute lymphoblastic leukemia remains one of the leading causes of death in children and young adults. Unlike acute lymphoblastic leukemia, acute myeloid leukemia is more common in older patients, with the peak incidence occurring at age 65, and 5-year survival varying from 10 to 90% depending on the molecular genetic subtype.

At the same time, 7-18% of deaths are associated with toxic effects of therapy. Currently, cytostatic chemotherapeutic drugs (cytarabine, ifosfamide, idarubicin, vincristine, doxorubicin, etc.) are used in the treatment of OL. These drugs are not selective in relation to tumor cells and have a toxic effect on all actively proliferating normal cells of the body.

As part of a study conducted by a group of authors to study the properties of Curaxin CBL0137, a new non-genotoxic compound with antitumor activity based on the drug's ability to non-covalently interact with DNA, causing translocation of the histone chaperone FACT to the chromatin fraction. As a result of the studies, the authors obtained new data on the drug that will be useful to a wide range of pharmacologists and practicing oncologists. I believe that the extreme importance of this study allows us to highly appreciate the work performed and recommend accepting this article for publication on a priority basis.

Reviewer 4 Report

Comments and Suggestions for Authors

The authors here demonstrate the synergistic antitumor effect of CBL0137 HAI and PD-1 blockade for cold hepatic metastases. The major mechanism is largely attributed to the depletion of myeloid-derived suppressor cells (MDSC) and activation of T cells in TME. This work is largely complete in the aspect of scientific conclusions and experimental methods, promoting the knowledge of how to evoke the immune response to cold hepatic metastases. I have to refer to some important points to be solved.

1.   The quality of WB images in Figure 2A, Figure 1B, Figure 6E, Figure S5A is extremely low and should be remade to ensure the publication. More seriously, there are two bands for the actin detection. Also, these band intensity of actin from different samples is also not consistent, which will interfere with the conclusion.

2.   The structure and the action of mode of CBL0137 is needed to be shown as Figure, although the previously reported literature is referred in the background.

3.   The Figure 2 is arranged in front of Figure 1.

4. All the Figures in the supplementary materials lack the caption description.

5.   The resolution of Figure 6 should be further improved. A wavy line under “sq” is shown in Figure 6E.

6.  The specificity of CBL0137 HAI to MDSC should be specially elucidated more clearly, although some explanations were provided. The model for CBL0137 HAI is needed to shown as a Figure.

7. The difference of antitumor effect for CBL0137 HAI and IV should be compared and explained in the discussion part.

8.  As shown in Figure 1A, CBL0137 induce 80% of cell death. How about the apoptosis rates for this compound against CT26 and 4T1? As shown in Figure S2A, CBL0137 HAI induce 10-15% of cell apoptosis. How about the expression level of PCNA as the proliferation biomarker for this compound against CT26 and 4T1? Seemingly, the cell apoptosis and proliferation biomarker are not consistent in showing the antitumor effect.

Round 2

Reviewer 1 Report

Comments and Suggestions for Authors

The authors have addressed my concerns. I have no further comment.

Reviewer 2 Report

Comments and Suggestions for Authors

acceptance in the current form

Reviewer 4 Report

Comments and Suggestions for Authors

acceptance of the revised manuscript for publication directly